# Practical device-independent quantum cryptography via entropy accumulation

Rotem Arnon-Friedman[1], Frédéric Dupuis[2,3], Omar Fawzi[4], Renato Renner[1] & Thomas Vidick[5]

Device-independent cryptography goes beyond conventional quantum cryptography by providing security that holds independently of the quality of the underlying physical devices. Device-independent protocols are based on the quantum phenomena of non-locality and the violation of Bell inequalities. This high level of security could so far only be established under conditions which are not achievable experimentally. Here we present a property of entropy, termed "entropy accumulation", which asserts that the total amount of entropy of a large system is the sum of its parts. We use this property to prove the security of cryptographic protocols, including device-independent quantum key distribution, while achieving essentially optimal parameters. Recent experimental progress, which enabled loophole-free Bell tests, suggests that the achieved parameters are technologically accessible. Our work hence provides the theoretical groundwork for experimental demonstrations of device-independent cryptography.

[1] Institute for Theoretical Physics, ETH-Zürich, Wolfgang-Pauli-Str. 27, 8093 Zürich, Switzerland. [2] Faculty of Informatics, Masaryk University, Brno, Czech Republic. [3] CNRS, LORIA, Université de Lorraine, Campus scientifique, BP 239, 54506 Vandoeuvre-lès-Nancy Cedex, France. [4] Laboratoire de l'Informatique du Parallélisme, LIP ENS de Lyon, 46 Allee d'Italie, 69364 Lyon Cedex 07, France. [5] Department of Computing and Mathematical Sciences, California Institute of Technology, 1200 E. California Blvd, Pasadena, CA 91125, USA. Correspondence and requests for materials should be addressed to R.A-F. (email: rotema@itp.phys.ethz.ch)

Device-independent (DI) quantum cryptographic protocols achieve an unprecedented level of security—with guarantees that hold (almost) irrespective of the quality, or trustworthiness, of the physical devices used to implement them[1]. The most challenging cryptographic task in which DI security has been considered is quantum key distribution (QKD); we will use this task as an example throughout the manuscript. In DIQKD, the goal of the honest parties, called Alice and Bob, is to create a shared key, unknown to everybody else but them. To execute the protocol, they hold a device consisting of two parts: each part belongs to one of the parties and is kept in their laboratories. Ideally, the device performs measurements on some entangled quantum states it contains.

In real life, the manufacturer of the device, called Eve, can have limited technological abilities (and hence cannot guarantee that the device's actions are exact and non-faulty) or even be malicious. The device itself is far too complex for Alice and Bob to open and assess whether it works as Eve alleges. Alice and Bob must therefore treat the device as a black box with which they can only interact according to the protocol. The protocol must allow them to test the possibly faulty or malicious device and decide whether using it to create their keys poses any security risk. The protocol guarantees that by interacting with the device according to the specified steps, the honest parties will either abort, if they detect a fault, or produce identical and secret keys (with high probability).

Adopting the DI approach is not only crucial for the paranoid cryptographers; even the most skilled experimentalist will recognise that a fully characterised, stable at all times, large-scale quantum device that implements a QKD protocol is extremely hard to build. Indeed, implementations of QKD protocols have been attacked by exploiting imperfections of the devices[2–5]. Instead of trying to come up with a "patch" each time an imperfection in the device is detected, DI protocols allow us to break the cycle of attacks and countermeasures.

The most important (in fact necessary) ingredient, which forms the basis of all DI protocols, is a "test for quantumness" based on the violation of a Bell inequality[6–9]. A Bell inequality[10,11] can be thought of as a game played by the honest parties using the device they share (Fig. 1). Different devices lead to different winning probabilities when playing the game. The game has a special "feature"—there exists a quantum device which achieves a winning probability $\omega_q$ greater than all classical, local, devices. Hence, if the honest parties observe that their device wins the game with probability $\omega_q$ they conclude that it must be non-local[11]. A recent

sequence of breakthrough experiments have verified the quantum advantage in such "Bell games" in a loophole-free way[12–14] (in particular, this means that the experiments were executed without making assumptions that could otherwise be exploited by Eve to compromise the security of a cryptographic protocol).

DI security relies on the following deep but well-established facts. High winning probability in a Bell game not only implies that the measured system is non-local, but more importantly that the kind of non-local correlations it exhibits cannot be shared: the higher the winning probability, the less information any eavesdropper can have about the devices' outcomes. The tradeoff between winning probability and secret randomness, or entropy, can be made quantitative[15,16].

The amount of entropy, or secrecy, generated in a single round of the protocol can therefore be calculated from the winning probability in a single game. The major challenge, however, consists in establishing that entropy accumulates additively throughout the multiple rounds of the protocol and use it to bound the total secret randomness produced by the device.

A commonly used assumption[17–21] to simplify this task is that the device held by the honest parties makes the same measurements on identical and independent quantum states in every round $i \in \{1, \ldots, n\}$ of the protocol. This implies that the device is initialised in some (unknown) state of the form $\sigma^{\otimes n}$, i.e., an independent and identically distributed (i.i.d.) state, and that the measurements have a similar structure. In that case, the total entropy created during the protocol can be easily related to the sum of the entropies generated in each round separately (as further explained below).

Unfortunately, although quite convenient for the analysis, the i.i.d. assumption cannot be justified a priori. When considering device-dependent protocols, such as the BB84 protocol[22], de Finetti theorems[23,24] can often be applied to reduce the task of proving the security in the most general case to that of proving security with the i.i.d. assumption. This approach was unsuccessful in the DI scenario, where known de Finetti theorems[23–26] do not apply. Hence, one cannot simply reduce a general security statement to the one proven under the i.i.d. assumption.

Without this assumption, however, very little is known about the structure of the untrusted device and hence also about its output. As a consequence, previous DIQKD security proofs had to address directly the most general case[27–29]. This led to security statements which are of limited relevance for practical experimental implementations; they are applicable only in an unrealistic regime of parameters, e.g., small amount of tolerable noise and large number of signals.

The work presented here resolves this situation. First, we provide a general information-theoretic tool that quantifies the amount of entropy accumulated during sequential processes which do not necessarily behave identically and independently in each step. We call this result the "Entropy Accumulation Theorem" (EAT). We then show how it can be applied to essentially reduce the problem of proving DI security in the most general case to that of the i.i.d. case. This allows us to establish simple and modular security proofs for DIQKD that yield tight key rates. Our quantitative results imply that the first proofs of principle experiments implementing a DIQKD protocol are within reach with today's state-of-the-art technology. Aside from its application to security proofs, the EAT can be used in other scenarios in quantum information such as the analysis of quantum random access codes.

| Alice: | Input | $x \in \{0, 1\}$ |
| | Output | $a \in \{0, 1\}$ |
| Bob: | Input | $y \in \{0, 1\}$ |
| | Output | $b \in \{0, 1\}$ |
| Win: | $a \oplus b = x \cdot y$ | |

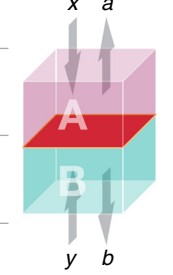

**Fig. 1** The Clauser–Horne–Shimony–Holt game[34]. Alice and Bob input bits, separately, into their parts of the shared device. Each part of the device supplies an output. The game is won if $a \oplus b = x \cdot y$. The optimal winning probability in this game for a classical device is 75%. A quantum device can get up to approximately 86% by measuring the maximally entangled state $|\Phi^+\rangle = (|00\rangle + |11\rangle)/\sqrt{2}$ with the following measurements: Alice's measurements $x = 0$ and $x = 1$ correspond to the Pauli operators $\sigma_z$ and $\sigma_x$, respectively, and Bob's measurements $y = 0$ and $y = 1$ to $(\sigma_z + \sigma_x)/\sqrt{2}$ and $(\sigma_z - \sigma_x)/\sqrt{2}$, respectively

## Results

In the following, we start by explaining the main steps in a security proof of DIQKD under the i.i.d. assumption using well-

established techniques. We then present the EAT and show how it can be used to extend the proof and achieve full security (i.e., without assuming an i.i.d. behaviour of the device).

**Security under the independent and identically distributed device assumption.** The central task when proving the security of cryptographic protocols consists in bounding the information that an adversary, called Eve, may obtain about certain values generated by the protocol, which are supposed to be secret. For QKD, the appropriate measure of knowledge, or rather uncertainty, is given by the smooth conditional min-entropy[30] $H_{\min}^{\varepsilon}(K|E)$, where $K$ is the raw data obtained by the honest parties, $E$ the quantum system held by Eve, and $\varepsilon$ a parameter describing the security of the protocol. The quantity $H_{\min}^{\varepsilon}(K|E)$ determines the maximal length of the secret key that can be created by the protocol. Hence, proving the security amounts to establishing a lower bound on $H_{\min}^{\varepsilon}(K|E)$. Evaluating $H_{\min}^{\varepsilon}(K|E)$ can be a daunting task, as the adversary's system $E$ is out of our control; in particular, it can have arbitrary dimension and share quantum correlations with the users' devices.

Most protocols consist of a basic building block, or "round", which is repeated a large number, $n$, of times; in each round $i$, the classical data $K_i$ is generated. The structure of a DIQKD protocol is shown in Box 1. The i.i.d. assumption means that the raw key $K_1^n = K_1, \ldots , K_n$ can be treated as a sequence of i.i.d. random variables $K_i$. That is, all the $K_i$ are identical and independent of one another. The eavesdropper has side information $E_i$ about each $K_i$. In this case, the total conditional min-entropy $H_{\min}^{\varepsilon}(K_1^n|E_1^n)$ can be directly related to the single-round conditional von Neumann entropy $H(K_i|E_i)$ using the quantum asymptotic equipartition property[31] (AEP), which asserts that

$$H_{\min}^{\varepsilon}\left(K_1^n\big|E_1^n\right) \geq nH(K_i|E_i) - c_{\varepsilon}\sqrt{n}, \qquad (1)$$

where $c_{\varepsilon}$ depends only on $\varepsilon$ (see the Methods section).

To get a bound on $H_{\min}^{\varepsilon}\left(K_1^n\big|E_1^n\right)$, we therefore need to analyse the secrecy, $H(K_i|E_i)$, resulting from a single round of the protocol. Depending on the considered scenario, a lower bound on $H(K_i|E_i)$ can be found using different techniques. For discrete- and continuous-variable QKD, for example, one can use the entropic uncertainty relations[32,33]. When dealing with DIQKD, a quantum advantage in a Bell game implies a lower-bound on $H(K_i|E_i)$ as discussed above.

The Clauser–Horne–Shimony–Holt (CHSH) game[34] (presented in Fig. 1) forms the basis for most DIQKD protocols. For this game, a tight bound on the secrecy as a function of the winning probability in the game was derived[19]. The bound implies that for any quantum state that wins the CHSH game with probability $\omega$, the entropy evaluated on the state of the system after the game has been played is at least

$$H(K_i|E_i) \geq 1 - h\left(\frac{1}{2} + \frac{1}{2}\sqrt{16\omega(\omega - 1) + 3}\right), \qquad (2)$$

where $h(\cdot)$ is the binary entropy function. This relation is shown in Fig. 2.

To compute the bound on $H(K_i|E_i)$, Alice and Bob need to collect the statistics they observe while running the protocol and estimate the winning probability $\omega$ appearing in Eq. (2); assuming the i.i.d. structure this is easily done using Hoeffding's inequality.

The conclusion of this section is the following. The i.i.d. assumption plays a crucial role in the above line of proof: it allows us to reduce the problem of calculating the total secrecy of the raw key created by the device to that of bounding the secrecy produced in one round. Instead of dealing with large-scale quantum systems, we are only required to understand the physics of small systems associated with just one round (as in Eq. (2)). The AEP appearing as Eq. (1) does the rest.

**Extending to full security.** Assuming the device behaves in an i.i.d. way goes completely against the DI setting by imposing a severe and even unrealistic restrictions on the implementation of the device. In particular, the assumption implies that the device does not include any, classical or quantum, internal memory (i.e.,

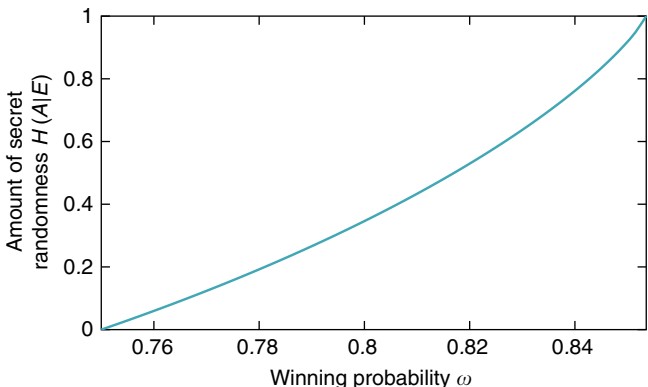

**Fig. 2** Secrecy for the Clauser–Horne–Shimony–Holt game vs. winning probability. The amount of secret randomness is quantified by the conditional von Neumann entropy $H(A|E)$. As soon as the winning probability is above the classical threshold of 75% some secret randomness is produced. The analytical bound[19] is stated as Eq. (2)

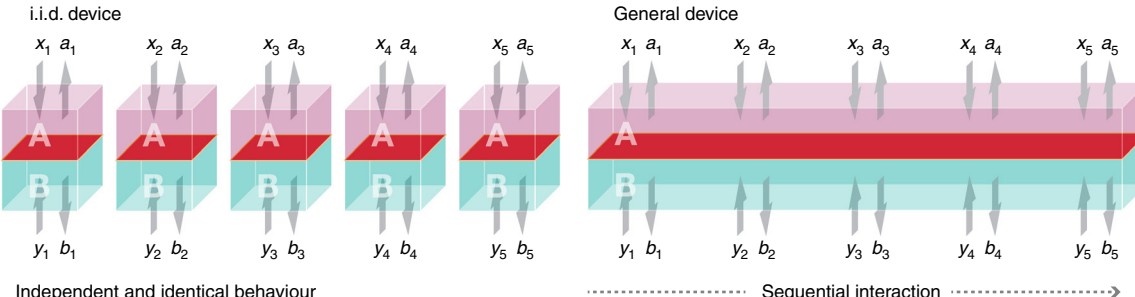

**Fig. 3** An independent and identically distributed device vs. a general one. An independent and identically distributed (i.i.d.) device (left) is initialised in some (unknown) i.i.d. state $\sigma^{\otimes n}$; each "small device" is described by one copy of the same bipartite state $\sigma$ and all copies are measured in the same way. A general device (right) is described by a bipartite quantum state $\rho$; in contrast to the i.i.d. case, any further division into subsystems is unknown. During the protocol, the state is measured through a sequential process: Alice and Bob use the device in the first round of the protocol and only then proceed to the second round, and so on

---

**Box 1 | Device-independent quantum key distribution protocol (simplified example)**

**Given:** A device for Alice and Bob that can play the chosen Bell game repeatedly

1. For every round $i \in [n]$ do Steps 2–4:
2. Alice and Bob choose $X_i, Y_i$ at random.
3. They input $X_i, Y_i$ to the device and record the outputs $A_i, B_i$.
4. Alice sets $K_i = A_i$

5. **Parameter estimation:** Alice and Bob estimate the average winning probability in the game from the observed data. If it is below the expected winning probability, $\omega_T$, they abort.

6. **Classical post processing:** Alice and Bob apply an error correction protocol and a privacy amplification protocol (both classical) on their raw keys $K$ and $B$.

---

its actions in one round cannot depend on the previous rounds) and cannot display time-dependent behaviour.

Our main contribution can be phrased as follows.

*Theorem* (*Security of DIQKD, informal*): Security of DIQKD in the most general case follows from security under the i.i.d. assumption. Moreover, the dependence of the key rate on the number of exchanged signals, $n$, is the same as the one in the i.i.d. case, up to terms that scale like $1/\sqrt{n}$. The key rates are plotted below.

We now explain the above theorem and how it is derived in more detail. A general device is described by an (unknown) tripartite state $\rho_{Q_A Q_B E}$, where the bipartite quantum state $\rho_{Q_A Q_B}$ is shared between Alice and Bob and $\rho_E$ belongs to Eve, together with the measurements applied to $\rho_{Q_A Q_B}$ when the device is used. No additional structure is assumed (see Fig. 3).

As mentioned above, the standard DIQKD protocol proceeds in rounds (recall Box 1): Alice and Bob use their components in the first round of the protocol and only then proceed to the second round, etc. We leverage this structure to bound the amount of entropy produced during a complete execution of the protocol.

To do so, we prove a generalisation of the AEP given in Eq. (1) to a scenario in which, instead of the raw key $K_1^n = K_1, \ldots, K_n$ being produced by an i.i.d. process, its parts $K_i$ are produced one after the other. In this case, each $K_i$ may depend not only on $i$-th round of the protocol but also on everything that happened in previous rounds (but not on the subsequent ones). We explain our tool, the EAT, in the following.

**The entropy accumulation theorem (EAT).** We describe here a simplified and informal version of the EAT, sufficient to understand how it can be used to prove the security of DI protocols; for the most general statements see the Methods section.

We consider processes which can be described by a sequence of $n$ maps $\mathcal{M}_1, \ldots, \mathcal{M}_n$, called "EAT channels", as shown in Fig. 4. Each $\mathcal{M}_i$ outputs two systems, $O_i$, which describes the information that should be kept secret, and $S_i$, describing some side information leaked by the map, together with a "memory" system $R_i$, which is passed on as an input to the next map $\mathcal{M}_{i+1}$. The systems $S_1^n$ describe the side information created during the process. A further quantum system, denoted by $E$, represents additional side information correlated to the initial state in the beginning of the considered process. The systems $O_1^n$ are then the ones in which entropy is accumulated conditioned on the side information $S_1^n$ and $E$.

To bound the entropy of $O_1^n$, we take into account global statistical properties. These are inferred by tests carried out by the protocol on a small sample of the generated outputs. To incorporate such statistical information, we consider for each round an additional classical value $C_i$ computed from $O_i$ and $S_i$.

Additionally, in each step of the process, the previous outcomes $O_1^{i-1}$ are independent of future information $S_i$ given all the past side information $S_1^{i-1}E$. By choosing $O_i$ and $S_i$ properly, this condition can be satisfied by sequential protocols such as DIQKD.

The EAT relates the total amount of entropy to the entropy accumulated in one step of the process. The latter is quantified by the minimal, or worst-case, von Neumann entropy produced by the maps $\mathcal{M}_i$ when acting on an input state that reproduces the correct statistics on $C_i$, i.e., states that satisfy $\mathcal{M}_i(\sigma)_{C_i} = \text{freq}(c_1^n)$ where $\text{freq}(c_1^n)$ is the empirical statistics, or frequency distribution, on $\mathcal{C}$ defined by $\text{freq}(c_1^n)(c) = \frac{|\{i \in \{1, \ldots, n\}: c_i = c\}|}{n}$.

To state the explicit result, we define a "min-tradeoff function", $f_{\min}$, from the set of probability distributions over $\mathcal{C}$ to the real numbers; $f_{\min}$ should be chosen as a convex differentiable function which is bounded above by the worst-case entropy just described:

$$f_{\min}\big(\text{freq}(c_1^n)\big) \leq H(O_i | S_i E). \tag{3}$$

An event $\Omega$ is defined by a subset of $\mathcal{C}^n$ and we write $p_\Omega = \sum_{c_1^n \in \Omega} (\rho_{O_1^n S_1^n E, C_1^n = c_1^n})$ for the probability of the event $\Omega$ and

$$\rho_{|\Omega} = \frac{1}{p_\Omega} \sum_{c_1^n \in \Omega} |c_1^n\rangle\langle c_1^n| \otimes \rho_{O_1^n S_1^n E, C_1^n = c_1^n} \tag{4}$$

for the state conditioned on $\Omega$. We further define a set $\hat{\Omega}$ over the set of frequencies such that for all $c_1^n$, $\text{freq}(c_1^n) \in \hat{\Omega}$ if and only if $c_1^n \in \Omega$.

*Theorem* (*EAT, informal*): For any EAT channels, an event $\Omega$ such that $\hat{\Omega}$ is a convex set, and a convex min-tradeoff function for which $f_{\min}\big(\text{freq}(c_1^n)\big) \geq t$ for any $\text{freq}(c_1^n) \in \hat{\Omega}$,

$$H_{\min}^\varepsilon\big(O_1^n | S_1^n E\big) > nt - \nu\sqrt{n}, \tag{5}$$

where the conditional smooth min-entropy is evaluated on $\rho_{|\Omega}$ and $\nu$ depends on the values $\|\nabla f_{\min}\|_\infty, \varepsilon, p_\Omega$, and the maximal dimension of the systems $O_i$.

Equation (5) asserts that, to first order in $n$, the total conditional smooth min-entropy is at least $n$ times the value of the min-tradeoff function, evaluated on the empirical statistics observed during the protocol (and hence linear in the number of rounds). In the special case where the EAT channels are independent and identical, the EAT is reduced to the quantum AEP; Eq. (5) is thus a generalisation of Eq. (1).

**DIQKD security via the EAT.** To gain intuition on how the EAT can be applied to DIQKD, note the following. Define the maps $\mathcal{M}_i$ to describe the joint behaviour of the honest parties and their respective uncharacterised device while playing a single round of a Bell game such as the CHSH game. Let $\Omega$ be the event of the protocol not aborting or a closely related event, e.g., the event that

the fraction of CHSH games won is above some threshold $\omega_T$. The state for which the smooth min-entropy is evaluated is $\rho_{|\Omega}$, i.e., the state at the end of the protocol conditioned on not aborting. This implies, in particular, a bound on $H_{\min}^{\varepsilon}(K_1^n|E)$.

Furthermore, the condition on the min-tradeoff function stated in Eq. (3) corresponds to the requirement that the distribution of $C_i$ equals $c_1^n$, which ensures that the entropy in Eq. (3) is evaluated on states that can be used to win the CHSH game with probability $\omega_T$. Thus, in order to devise an appropriate min-tradeoff function, we can use the relation appearing in Eq. (2); the exact details are given in the Methods section. This results in a tight bound on the amount of entropy created in each step of the protocol. In this sense, we reduce the problem of proving the security of the whole protocol to that of a single round.

Using the EAT we get a bound on $H_{\min}^{\varepsilon}(K_1^n|E)$ which, to first order in $n$, coincides with the one derived under the i.i.d. assumption and is thus optimal. The final key rate $r = \ell/n$ (where $\ell$ is the length of a key) produced in a DIQKD protocol depends on this amount of entropy and the amount of information leaked during standard classical post-processing steps. We plot the results for specific choices of parameters in Fig. 5.

To calculate the key rate, one must have some honest implementation of the protocol in mind; this is given by what the experimentalists think (or guess) is happening in their experiment when an adversary is not present. It does not, in any way, restrict the actions of the adversary or the types of imperfections in the device. We consider the following honest

implementation, but the analysis can be adapted to any other implementation of interest.

In the realisation of the device, in each round, Alice and Bob share the two-qubit Werner state $\rho_{Q_A Q_B} = (1 - \nu)|\Phi^+\rangle\langle\Phi^+| + \nu\mathbb{1}/4$ resulting from a depolarisation noise acting on the maximally entangled state $|\Phi^+\rangle$. In every round, the measurements for $X_i, Y_i \in \{0, 1\}$ are as described in Fig. 1 and for $Y_i = 2$ Bob's measurement is $\sigma_z$. The winning probability in the CHSH game (restricted to $X_i, Y_i \in \{0, 1\}$) using these measurements on $\rho_{Q_A Q_B}$ is $\omega_{\exp} = [2 + \sqrt{2}(1 - \nu)]/4$. The quantum bit error rate $Q = \Pr[A_i \neq B_i|(X_i, Y_i) = (0, 2)]$ for the above state and measurements is given by $Q = \nu/2$.

The key rate $r$ is plotted in Fig. 5. For $n = 10^{15}$, the curve essentially coincides with the rate achieved in the asymptotic i.i.d. case[19]. Since the latter was shown to be optimal[19], it provides an upper bound on the key rate and the amount of tolerable noise. Hence, for large enough $n$ our rates become optimal and the protocol can tolerate up to the maximal error rate $Q = 7.1\%$. For comparison, the previously established explicit rates[28] are well below the lowest curve presented in Fig. 5, even when the number of signals goes to infinity, with a maximal noise tolerance of 1.6%. Moreover, our key rates are comparable to those achieved in device-dependent QKD protocols[35].

## Discussion

The information theoretic tool, the EAT, reveals a novel property of entropy: the operationally relevant total uncertainty about an $n$-partite system created in a sequential process corresponds to the sum of the entropies of its parts, even without an independence assumption.

Using the EAT, we show that practical and realistic protocols can be used to achieve the unprecedented level of DI security. The next major challenge in experimental implementations is a field demonstration of a DIQKD protocol. This would provide the strongest cryptographic experiment ever realised. The work presented here provides the theoretical groundwork for such experiments. Our quantitative results imply that the first proofs of principle experiments, with small distances and small rates, are within reach with today's state-of-the-art technology, which recently enabled the violation of Bell inequalities in a loophole-free way.

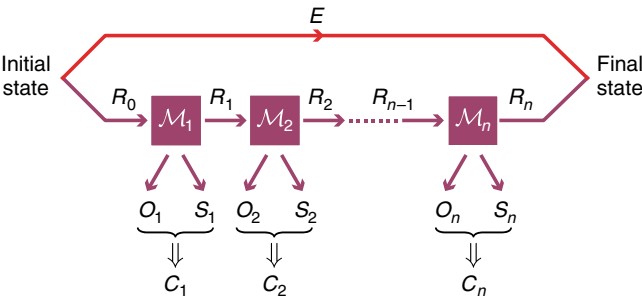

**Fig. 4** Sequential processes. Each map in the sequence $\mathcal{M}_i$ outputs $O_i$, which describes the information that should be kept secret, and $S_i$, describing some side information leaked by the map, together with a "memory" system $R_i$, which gets passed on to the next map $\mathcal{M}_{i+1}$. In each step, an additional classical value $C_i$ is calculated from $O_i$ and $S_i$

## Methods

We state here the main theorems of our work and sketch the proofs. Using the explicit expressions given below, one can reproduce the key rates presented in Fig. 5.

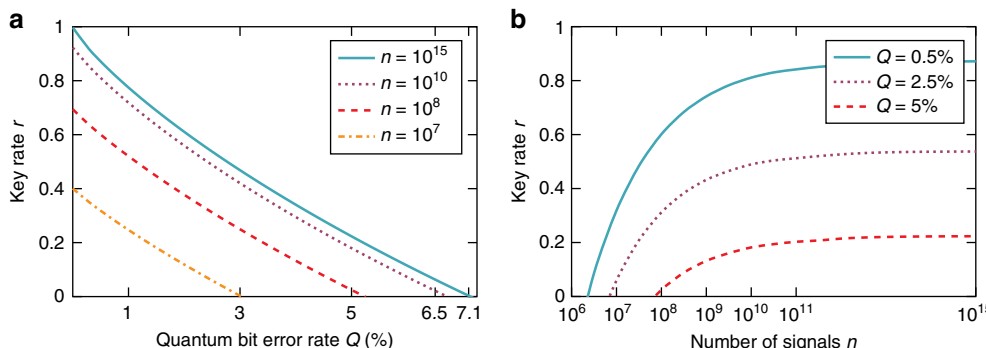

**Fig. 5** Key rate in a DIQKD protocol. The plots show the key rate $r$ as a function of **a** the quantum bit error rate $Q$ and **b** the number of signals $n$. The completeness error, i.e., the probability that the protocol aborts when using an honest implementation of the device, e.g., due to statistical fluctuations, was chosen to be $\varepsilon_{QKD}^c = 10^{-2}$. The soundness error, which quantifies the maximum tolerated deviation of the actual protocol from a hypothetical one where a perfectly random and completely secret key is produced for Alice and Bob, is taken to be $\varepsilon_{QKD}^s = 10^{-5}$. Both of these values are considered to be realistic and relevant for actual applications. The rates are calculated using Eq. (35) which is derived in the Methods section

**The formal statement and proof idea of the EAT**. In this section, we are interested in the general question of whether entropy accumulates, in the sense that the operationally relevant total uncertainty about an $n$-partite system $O_1^n$ corresponds to the sum of the entropies of its parts $O_i$. The AEP, given in Eq. (1), implies that this is indeed the case to first order in $n$—under the assumption that the parts $O_i$ are identical and independent of each other. Our result shows that entropy accumulation occurs for more general processes, i.e., without an independence assumption, provided one quantifies the uncertainty about the individual systems $O_i$ by the von Neumann entropy of a suitably chosen state.

The type of processes that we consider are those that can be described by a sequence of channels, as illustrated in Fig. 4. Such channels are called EAT channels and are formally defined as follows.

*Definition 1 (EAT channels)*: EAT channels $\mathcal{M}_i : R_{i-1} \rightarrow R_i O_i S_i C_i$, for $i \in [n]$, are CPTP (completely positive trace preserving) maps such that for all $i \in [n]$:

1. $C_i$ are finite-dimensional classical systems (random variables). $O_i$, $S_i$, and $R_i$ are quantum registers. The dimension of $O_i$ is $d_{O_i}$.
2. For any input state $\sigma_{R_{i-1}R'}$, where $R'$ is a register isomorphic to $R_{i-1}$, the output state $\sigma_{R_i O_i S_i C_i R'} = (\mathcal{M}_i \otimes \mathbb{I}_{R'})(\sigma_{R_{i-1}R'})$ has the property that the classical value $C_i$ can be measured from the system $\sigma_{O_i S_i}$ without changing the state.
3. For any initial state $\rho_{R_0 E}^0$, the final state $\rho_{O_1^n S_1^n C_1^n E} = ((\text{Tr}_{R_n} \circ \mathcal{M}_n \circ \ldots \circ \mathcal{M}_1) \otimes \mathbb{I}_E)\rho_{R_0 E}^0$ fulfils the Markov chain condition $O_1^{i-1} \leftrightarrow S_1^{i-1}E \leftrightarrow S_i$ for each $i \in [n]$.

In the above definition, $O_1^{i-1} \leftrightarrow S_1^{i-1}E \leftrightarrow S_i$ if and only if their conditional mutual information is 0, $I(O_1^{i-1} : S_i | S_1^{i-1}E) = 0$.

Next, one should find an adequate way to quantify the amount of entropy which is accumulated in a single step of the process, i.e., in an application of just one channel. To do so, let $p$ be a probability distribution over $\mathcal{C}$, where $\mathcal{C}$ denotes the common alphabet of $C_1, \ldots, C_n$, and $R'$ be a system isomorphic to $R_{i-1}$. We define the set of states

$$\Sigma_i(p) = \{\sigma_{O_i S_i C_i R_i R'} = (\mathcal{M}_i \otimes \mathbb{I}_{R'})(\tau_{R_{i-1}R'}) : \tau \in \text{D}(R_{i-1} \otimes R') \text{ and } \sigma_{C_i} = p\}, \quad (6)$$

where $\sigma_{C_i}$ denotes the probability distribution over $\mathcal{C}$ with the probabilities given by $\langle c | \sigma_{C_i} | c \rangle$.

The tradeoff functions for the EAT channels are defined below.

*Definition 2 (Tradeoff functions)*: A real function is called a min- or max-tradeoff function for $\mathcal{M}_i$ if it satisfies

$$f_{\min}(p) \leq \inf_{\sigma \in \Sigma_i(p)} H(O_i | S_i R')_\sigma \quad (7)$$

or

$$f_{\max}(p) \geq \sup_{\sigma \in \Sigma_i(p)} H(O_i | S_i R')_\sigma, \quad (8)$$

respectively, and if it is convex or concave, respectively. If the set $\Sigma_i(p)$ is empty, then the infimum and supremum are by definition equal to $\infty$ and $-\infty$, respectively, so that the conditions are trivial.

To get some intuition as to why the above definition is the "correct" one, consider the following classical example. Each EAT channel outputs a single bit $O_i$ without any side information $S_i$ about it; the system $E$ is empty as well. Every bit can depend on the ones produced previously. We would like to extract randomness out of the sequence $O_1^n$; for this we should find a lower bound on $H_{\min}^\varepsilon(O_1^n)$.

We ask the following question—given the randomness of $O_1$ which is already accounted for, how much randomness does $O_2$ contribute? One possible guess is the conditional von Neumann entropy $H(O_2 | O_1) = \mathbb{E}_{o_1, o_2}[-\log \Pr(o_2 | o_1)]$. If, however, $O_1$ is uniform while $O_2$ is fixed when $O_1 = 0$ and uniform otherwise, then $H(O_2 | O_1)$ is too optimistic; the amount of extractable randomness is quantified by the smooth min-entropy, which depends on the most probable value of $O_1 O_2$, and not by an average quantity as the von Neumann entropy.

Another possible guess is a worst-case version of the min-entropy $H_{\min}^{\text{w.c.}} = \min_{o_1, o_2}[-\log \Pr(o_2 | o_1)]$. This, however, is too pessimistic; when the $O_i$'s are independent of each other, the extractable amount of randomness behaves like the von Neumann entropy in first order, and not like the min-entropy.

We therefore choose an intermediate quantity— $\min_{o_1} \mathbb{E}_{o_2}[-\log \Pr(o_2 | o_1)] = \min_{o_1} H(O_2 | O_1 = o_1)$. That is, this quantity is the von Neumann entropy of $O_2$, evaluated for the worst-case state in the beginning of the second step of the process. The min-tradeoff function defined above is the quantum analogue version of this.

Informally, the min-tradeoff function can be understood as the amount of entropy available from a single round, conditioned on the outputs of the previous rounds. Since we condition on the previous rounds, one can think of the randomness of the current round as independent from past events. Intuitively, this suggests that, by appropriately generalising the proof of the AEP, one can argue that the entropy that is contributed by this independent randomness in each round accumulates.

The formal statement of the EAT is as follows.

*Theorem 3 (EAT, formal)*: Let $\mathcal{M}_i : R_{i-1} \rightarrow R_i O_i S_i C_i$ for $i \in [n]$ be EAT channels, $\rho$ be the final state, $\Omega$ an event defined over $\mathcal{C}^n$, $p_\Omega$ the probability of $\Omega$ in $\rho$, and $\rho_{|\Omega}$ the final state conditioned on $\Omega$. Let $\varepsilon_s \in (0, 1)$.

For $f_{\min}$, a min-tradeoff function for $\{\mathcal{M}_i\}$, $\Omega = \{\text{freq}(c_1^n) | c_1^n \in \Omega\}$ convex, and any $t \in \mathbb{R}$ such that $f_{\min}(\text{freq}(c_1^n)) \geq t$ for any $c_1^n \in \mathcal{C}^n$ for which $\Pr[c_1^n]_{\rho_{|\Omega}} > 0$,

$$H_{\min}^\varepsilon(O_1^n | S_1^n E)_{\rho_{|\Omega}} > nt - \nu\sqrt{n}, \quad (9)$$

where

$$\nu = 2\left(\log(1 + 2d_{O_i}) + \left\lceil \|\nabla f_{\min}\|_\infty \right\rceil\right)\sqrt{1 - 2\log(\varepsilon_s \cdot p_\Omega)}. \quad (10)$$

Similarly, for $f_{\max}$ a max-tradeoff function and $t \in \mathbb{R}$ such that $f_{\max}(\text{freq}(c_1^n)) \leq t$ for any $\mathbf{c} \in \mathcal{C}^n$ for which $\Pr[\mathbf{c}]_{\rho_{|\Omega}} > 0$,

$$H_{\max}^\varepsilon(O_1^n | S_1^n E)_{\rho_{|\Omega}} < nt - \nu\sqrt{n}. \quad (11)$$

The two most important properties of the above statement are that the first-order term is linear in $n$ and that $t$ is the von Neumann entropy of a suitable state (as explained above). This implies that the EAT is tight to first order in $n$.

We remark that the Markov chain conditions are important, in the sense that dropping them completely would render the statement invalid.

We now give a rough proof sketch of the $H_{\min}^\varepsilon$ case; the bound for $H_{\max}^\varepsilon$ follows from an almost identical argument. The proof has a similar structure to that of the quantum AEP[31], which we can retrieve as a special case. The proof relies heavily on the "sandwiched" Rényi entropies[36,37], which is a family of entropies that we will denote here by $H_\alpha$, where $\alpha$ is a real parameter ranging from $\frac{1}{2}$ to $\infty$, and which corresponds to the max-entropy at $\alpha = \frac{1}{2}$, to the von Neumann entropy when $\alpha = 1$, and to the min-entropy when $\alpha = \infty$.

The basic idea is to first lower bound the $H_{\min}^\varepsilon$ term by $H_\alpha$ using the following general bound[31,38–40]:

$$H_{\min}^\varepsilon(A|B) > H_\alpha(A|B) - \frac{1}{\alpha - 1}O(\log(1/\varepsilon)). \quad (12)$$

Then, we lower bound the $H_\alpha$ term by the von Neumann entropy using the following[31,39]:

$$H_\alpha(A|B) > H(A|B) - (\alpha - 1)O\left((\log d_A)^2\right). \quad (13)$$

Now, we could simply chain these two inequalities and apply them to $H_{\min}^\varepsilon(O_1^n | S_1^n E)$. However, this would result in a very poor bound due to the $O\left((\log d_A)^2\right)$ term in Eq. (13), which in our case would be $O(n^2)$. To get the bound we want, we need to reduce this term to $O(n)$; choosing $\alpha \approx 1 + \frac{1}{\sqrt{n}}$ would then produce a bound with the right scaling.

The trick we use to achieve this is to decompose $H_\alpha(O_1^n | S_1^n E)$ into $n$ terms of constant size before applying Eq. (13). In the quantum AEP[31], this step is immediate since the state is i.i.d. Here, we must use more sophisticated techniques. Specifically, we use the following chain rule for the sandwiched Rényi entropy to decompose $H_\alpha(O_1^n | S_1^n E)$ into $n$ terms:

*Theorem 4*: Let $\rho_{RA_1 B_1}^0$ be a density operator on $R \otimes A_1 \otimes B_1$ and $\mathcal{M} = \mathcal{M}_{A_2 B_2 \leftarrow R}$ be a CPTP map. Assuming that $\rho_{A_1 B_1 A_2 B_2} = \mathcal{M}(\rho_{RA_1 B_1}^0)$ satisfies the Markov condition $A_1 \leftrightarrow B_1 \leftrightarrow B_2$, we have

$$H_\alpha(A_1 | B_1)_\rho + \inf_\omega H_\alpha(A_2 | B_2 A_1 B_1)_{\mathcal{M}(\omega)} \leq H_\alpha(A_1 A_2 | B_1 B_2)_\rho, \quad (14)$$

where the infimum ranges over density operators $\omega_{RA_1 B_1}$ on $R \otimes A_1 \otimes B_1$. Moreover, if $\rho_{RA_1 B_1}^0$ is pure, then we can optimise over pure states $\omega_{RA_1 B_1}$.

Implementing this proof strategy then yields the following chain of inequalities:

$$
\begin{aligned}
& H_{\min}^\varepsilon(O_1^n | S_1^n E)_\rho \\
& > H_\alpha(O_1^n | S_1^n E)_\rho - \frac{1}{\alpha - 1}O(\log(1/\varepsilon)) \\
& \geq \sum_i \inf_{\omega_{R'R_i}} H_\alpha(O_i | S_i R')_{\mathcal{M}_i(\omega)} - \frac{1}{\alpha - 1}O(\log(1/\varepsilon)) \\
& > \sum_i \inf_\omega H(O_i | S_i R')_{\mathcal{M}_i(\omega)} - \frac{1}{\alpha - 1}O(\log(1/\varepsilon)) \\
& \quad - n(\alpha - 1)O\left((\log d_{O_i})^2\right) \\
& \geq \sum_i \inf_\omega H(O_i | S_i R')_{\mathcal{M}_i(\omega)} - O(\sqrt{n}).
\end{aligned}
\quad (15)
$$

However, this does not yet take into account the sampling over the $C_i$ subsystems. To do this, we tweak the EAT channels $\mathcal{M}_i$ to output two extra systems $D_i$ and $\overline{D}_i$ which contain an amount of entropy that depends on the value of $C_i$ observed. To define this, let $g$ be an affine lower bound on $f_{\min}$, let $[g_{\min}, g_{\max}]$ be the smallest interval that contains the range of $g$, and set $\bar{g} := \frac{1}{2}(g_{\min} + g_{\max})$. Then, we define $\mathcal{D}_i : C_i \rightarrow C_i D_i \overline{D}_i$ as

$$\mathcal{D}_i(X) = \sum_c \langle c | X | c \rangle \cdot | c \rangle \langle c |_{C_i} \otimes \tau(c)_{D_i \overline{D}_i}, \quad (16)$$

---

**Box 2 | Entropy accumulation protocol (based on the CHSH game)**

**Given:**

$D$—device that can play the CHSH game repeatedly

$m \in \mathbb{N}_+$—number of blocks

$s_{\max} \in \mathbb{N}_+$—maximal length of a block

$\gamma \in (0, 1]$—probability of a test round

$\omega_{\exp}$—expected winning probability in the honest implementation

$\delta_{\mathrm{est}} \in (0, 1)$—width of the statistical confidence interval

1.  For every block $j \in [m]$ do Steps 2-10:
2. Set $i = 0$ and $\tilde{C}_j = \perp$.
3. If $i \leq s_{\max}$:
4. Set $i = i + 1$.
5. Alice and Bob choose $T_i \in \{0, 1\}$ at random such that $\Pr(T_i = 1) = \gamma$.
6. If $T_i = 1$ Alice and Bob choose inputs $X_i \in \{0,1\}$ and $Y_i \in \{0,1\}$.
7. If $T_i = 0$ they choose inputs $X_i \in \{0, 1\}$ and $Y_i = 2$.
8. Alice and Bob use $D$ with $X_i, Y_i$ and record their outputs as $A_i, B_i$.
9. If $T_i = 0$ Bob updates $B_i$ to $B_i = \perp$.
10. If $T_i = 1$ they set $\tilde{C}_j = w(A_i, B_i, X_i, Y_i)$ and $i = s_{\max} + 1$.
11. Alice and Bob abort if $\sum_{j \in [m]} \tilde{C}_j < [\omega_{\exp}(1 - (1 - \gamma)^{s_{\max}}) - \delta_{\mathrm{est}}] \cdot m$.

---

where $\tau(c)$ is a mixture between a maximally entangled state and a fully mixed state such that the marginal on $\overline{D}_i$ is uniform, and such that $H_\alpha\left(D_i | \overline{D}_i\right)_{\tau(c)} = \overline{g} - g(\delta_c)$ (here $\delta_c$ stands for the distribution with all the weight on element $c$). To ensure that this is possible, we need to choose $d_{D_i}$ large enough, and it turns out that setting $d_{D_i} = \lceil 2^{\|\nabla g\|_\infty} \rceil$ suffices. We can then define a new sequence of EAT channels by $\mathcal{M}_i = \mathcal{D}_i \circ \mathcal{M}_i$.

Armed with this, we apply the above argument to our new EAT channels. On the one hand, a more sophisticated version of Eq. (12) yields:

$$H_{\min}^{\varepsilon}\left(O_1^n | S_1^n E\right)_{\rho_{|\Omega}} \geq H_\alpha\left(O_1^n D_1^n | S_1^n E \overline{D}_1^n\right)_\rho \\ - n\overline{g} + nt - O(\sqrt{n})\log\left(\tfrac{1}{p_\Omega}\right) \quad (17)$$

On the other hand, the argument from Eq. (15) can be used here to give

$$H_\alpha\left(O_1^n D_1^n | S_1^n E \overline{D}_1^n\right)_\rho > n\overline{g} - O(\sqrt{n})\left(\log(d_{O_i} d_{D_i})^2\right). \quad (18)$$

Combining these two bounds then yields the theorem.

We remark that some of the concepts used in this work generalise techniques proposed in the recent security proofs for DI cryptography[29].

**Entropy accumulation protocol.** To analyse the key rates of the DIQKD protocol, we first find a lower bound on the amount of entropy accumulated during the run of the protocol, when the honest parties use their device to play the Bell games repeatedly. To this end, we consider the "entropy accumulation protocol" shown in Box 2. This protocol can be seen as the main building block of many DI cryptographic protocols.

The entropy accumulation protocol creates $m$ blocks of bits, each of maximal length $s_{\max}$. Each block ends (with high probability) with a test round; this is a round in which Alice and Bob play the CHSH game with their device so that they can verify that the device acts as expected. The probability of each round to be a test round is $\gamma$. The rest of the rounds are generation rounds, in which Bob chooses a special input for his component of the device. In the end of the protocol, Alice and Bob check whether they had sufficiently many test rounds in which they won the CHSH game. If not, they abort.

We note that the protocol is complete, in the sense that there exists an honest implementation of it (possibly noisy) which does not abort with high probability. Denoting the completeness error, i.e., the probability that the protocol aborts for an honest implementation of the devices $D$, by $\varepsilon_{EA}^c$, one can easily show using Hoeffding's inequality that for an honest i.i.d. implementation $\varepsilon_{EA}^c \leq \exp\left(-2n\delta_{\mathrm{est}}^2\right)$.

Next, we show that the protocol is also sound. That is, for any device $D$, if the probability that the protocol does not abort is not too small, then the total amount of smooth min-entropy is sufficiently high.

The EAT can be used to bound the total amount of smooth min-entropy, $H_{\min}^{\varepsilon_s}\left(A_1^n B_1^n | X_1^n Y_1^n T_1^n E\right)_{\rho_{|\Omega}}$, created when running the entropy accumulation protocol, given that it did not abort. Here $n$ denotes the expected number of rounds of the protocol and $\varepsilon_s$ is one of the security parameters (to be fixed later).

Below we use the following notation. For each block $j \in [m]$, $\vec{A}_j$ denotes the string that includes Alice's outputs in block $j$ (note that the length of this string is unknown, but it is at most $s_{\max}$). $\vec{B}_j$, $\vec{X}_j$, and $\vec{Y}_j$ are defined analogously. To use the EAT, we make the following choices of random variables:

$$O_i \to \vec{A}_j \vec{B}_j \quad (19)$$

$$S_i \to \vec{X}_j \vec{Y}_j \vec{T}_j \quad (20)$$

$$C_i \to \tilde{C}_j \quad (21)$$

$$E \to E. \quad (22)$$

The event $\Omega$ is the event of not aborting the protocol, as given in Step 11 in Box 2:

$$\Omega = \left\{ \tilde{C}_1^m \mid \sum_{j | \tilde{C}_j = 1} 1 < \left[\omega_{\exp}(1 - (1 - \gamma)^{s_{\max}}) - \delta_{\mathrm{est}}\right] \cdot m \right\}. \quad (23)$$

The EAT channels are chosen to be

$$\mathcal{M}_j : R_{j-1} \to R_j \vec{A}_j \vec{B}_j \vec{X}_j \vec{Y}_j \tilde{C}_j, \quad (24)$$

where $\mathcal{M}_j$ describes Steps 2–10 of block $j$ in the entropy accumulation protocol (Box 2). These channels include both the actions made by Alice and Bob as well as the operations made by the device $D$ in these steps. Note that the Device's operations can always be described within the formalism of quantum mechanics, although we do not assume we know them. The registers $R_{j-1}$ and $R_j$ hold the quantum state of the device in the beginning and the end of the $j$'th step of the protocol, respectively.

*Lemma 5:* The channels $\mathcal{M}_j$ described above are EAT channels.

*Proof.* For the channels to be EAT channels, they need to fulfil the conditions given in Definition 1. We show that this is indeed the case. First, $\tilde{C}_j$ are classical registers with $\tilde{C}_j \in \{0, 1, \perp\}$ and $d_{\vec{A}_j} \times d_{\vec{B}_j} \leq 6^{s_{\max}}$. Second, $\tilde{C}_j$ is determined by the classical registers $\vec{A}_j$, $\vec{B}_j$, $\vec{X}_j$, $\vec{Y}_j$, $\vec{T}_j$ as shown in Box 2. Therefore, $\tilde{C}_j$ can be calculated without modifying the marginal on those registers. The third condition is also fulfilled since the inputs are chosen independently in each round and hence $\vec{A}_{1\ldots j-1}\vec{B}_{1\ldots j-1} \leftrightarrow \vec{X}_{1\ldots j-1}\vec{Y}_{1\ldots j-1}\vec{T}_{1\ldots j-1}E \leftrightarrow \vec{X}_j \vec{Y}_j \vec{T}_j$ trivially holds.

To continue one should devise a min-tradeoff function. Let $\tilde{p}$ be the probability distribution describing $\tilde{C}_j$. We remark that due to the structure of our EAT channels, it is sufficient to consider $\tilde{p}$ for which $\tilde{p}(1) + \tilde{p}(0) = 1 - (1 - \gamma)^{s_{\max}}$ (otherwise the set $\Sigma$ defined in Eq. (6) is an empty set).

The following lemma gives a lower bound on the von Neumann entropy of the outputs in a single block.

*Lemma 6:* Let $\overline{s} = \frac{1 - (1 - \gamma)^{s_{\max}}}{\gamma}$ be the expected length of a block and $h$ the binary entropy function. Then,

$$H\left(\vec{A}_j \vec{B}_j | \vec{X}_j \vec{Y}_j \vec{T}_j R'\right) \geq \overline{s}\left[1 - h\left(\frac{1}{2} + \frac{1}{2}\sqrt{16\omega^*(\omega^* - 1) + 3}\right)\right], \quad (25)$$

where the entropy is evaluated on a state that wins the CHSH game, in the test

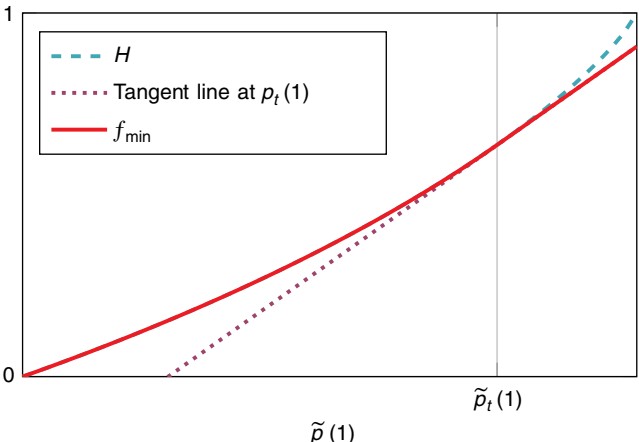

**Fig. 6** The construction of the min-tradeoff function $f_{min}$. The plot shows the values of the min-tradeoff function on a slice $\tilde{p}(0) + \tilde{p}(1) = 1 - (1 - \gamma)^{s_{max}}$

round, with probability

$$\omega^* = \frac{\tilde{p}(1)}{1 - (1 - \gamma)^{s_{max}}}. \tag{26}$$

*Proof sketch.* The amount of entropy accumulated in a single round in a block is given in Eq. (2) in the main text. To get the amount of entropy accumulated in a block, one can use the chain rule for the von Neumann entropy. The result is then

$$H\left(\vec{A}_j \vec{B}_j | \vec{X}_j \vec{Y}_j \vec{T}_j R'\right)$$
$$\geq \sum_{i \in [s_{max}]} (1 - \gamma)^{(i-1)} \left[1 - h\left(\tfrac{1}{2} + \tfrac{1}{2}\sqrt{16\omega_i(\omega_i - 1) + 3}\right)\right], \tag{27}$$

where the pre-factors $(1 - \gamma)^{(i-1)}$ are attributed to the fact that the entropy in each round is non-zero only if the round is part of the block, i.e., if a test round was not performed before the $i$'th round in the block, and $\omega_i$ denotes the winning probability in the $i$'th round (given that a test was not performed before).

The value of each $\omega_i$ is not fixed completely given $\omega^*$. However, by the operation of the EAT channels the following relation holds:

$$\omega^*(1 - (1 - \gamma)^{s_{max}}) = \sum_{i \in [s_{max}]} \gamma(1 - \gamma)^{(i-1)}\omega_i. \tag{28}$$

To conclude the proof, we thus need to minimise $H\left(\vec{A}_j \vec{B}_j | \vec{X}_j \vec{Y}_j \vec{T}_j R'\right)$ under the above constraint. Using standard techniques, e.g., Lagrange multipliers, one can see that the minimal value of this entropy is achieved for $\omega_i = \omega^*$ for all $i$ and the lemma follows.

The bound given in the above lemma can now be used to define the min-tradeoff function $f_{min}(\tilde{p})$. As the derivative of the function plays a role in the final bound, we must make sure it is not too large at any point. This can be enforced by "cutting" the function at a chosen point $\tilde{p}_t$ and "gluing" it to a linear function starting at that point, as shown in Fig. 6. $\tilde{p}_t$ can be chosen depending on the other parameters such that the total amount of smooth min-entropy is maximal. Following this idea, the resulting min-tradeoff function is given by

$$g(\tilde{p}) =$$
$$\begin{cases} \bar{s}\left[1 - h\left(\tfrac{1}{2} + \tfrac{1}{2}\sqrt{16\tfrac{\tilde{p}(1)}{1-(1-\gamma)^{s_{max}}}\left(\tfrac{\tilde{p}(1)}{1-(1-\gamma)^{s_{max}}} - 1\right) + 3}\right)\right] & \tfrac{\tilde{p}(1)}{1-(1-\gamma)^{s_{max}}} \in \left[0, \tfrac{2+\sqrt{2}}{4}\right] \\ \bar{s} & \tfrac{\tilde{p}(1)}{1-(1-\gamma)^{s_{max}}} \in \left[\tfrac{2+\sqrt{2}}{4}, 1\right], \end{cases} \tag{29}$$

$$f_{min}(\tilde{p}, \tilde{p}_t) = \begin{cases} g(\tilde{p}) & \tilde{p}(1) \leq \tilde{p}_t(1) \\ \frac{d}{d\tilde{p}(1)}g(\tilde{p})\Big|_{\tilde{p}_t} \cdot \tilde{p}(1) + \left(g(\tilde{p}_t) - \frac{d}{d\tilde{p}(1)}g(\tilde{p})\Big|_{\tilde{p}_t} \cdot \tilde{p}_t(1)\right) & \tilde{p}(1) > \tilde{p}_t(1). \end{cases} \tag{30}$$

Let $\varepsilon_{EA}$ be the desired error probability of the entropy accumulation protocol. We can then use Theorem 3 to say that either the probability of the protocol aborting is greater than $1 - \varepsilon_{EA}$ or the following bound on the total smooth min-entropy holds:

$$H_{min}^{\varepsilon_s}\left(A_1^n B_1^n | X_1^n Y_1^n T_1^n E\right)_{\rho_{|\Omega}} > m \cdot \eta_{opt}(\varepsilon_s, \varepsilon_{EA})$$
$$= \tfrac{n}{s} \cdot \eta_{opt}(\varepsilon_s, \varepsilon_{EA}), \tag{31}$$

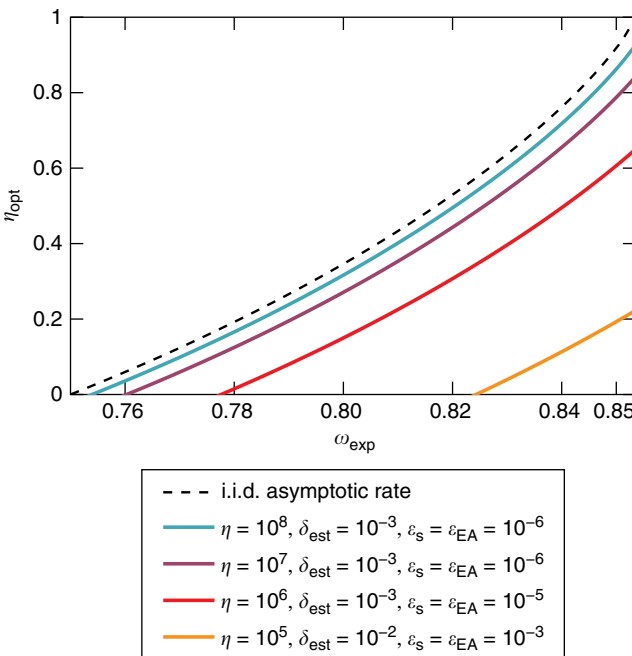

**Fig. 7** Entropy rate for entropy accumulation protocol. $\eta_{opt}(\omega_{exp})$ for $\gamma = 1$, $s_{max} = 1$ and several choices of $\delta_{est}$, $n$, $\varepsilon_{EA}$, and $\varepsilon_s$. We optimise the rates over all parameters which are not explicitly stated in the figure. The dashed line shows the optimal asymptotic ($n \to \infty$) rate under the assumption that the devices are such that Alice, Bob, and Eve share an (unknown) i.i.d. state and $n \to \infty$

where

$$\eta(\tilde{p}, \tilde{p}_t, \varepsilon_s, \varepsilon_e) = f_{min}(\tilde{p}, \tilde{p}_t) - \tfrac{1}{\sqrt{m}}2(\log(1 + 2 \cdot 6^{s_{max}})$$
$$+ \left\|\tfrac{d}{d\tilde{p}(1)}g(\tilde{p})\right\|_\infty)\sqrt{1 - 2\log(\varepsilon_s \cdot \varepsilon_e)},$$
$$\eta_{opt}(\varepsilon_s, \varepsilon_e) = \max_{\tfrac{3}{4} < \tilde{p}_t(1) < \tfrac{2+\sqrt{2}}{4}} \eta\left(\omega_{exp}(1 - (1 - \gamma)^{s_{max}}) - \delta_{est}, \tilde{p}_t, \varepsilon_s, \varepsilon_e\right). \tag{32}$$

To illustrate the behaviour of the entropy rate $\eta_{opt}$, we plot it as a function of the expected Bell violation $\omega_{exp}$ in Fig. 7 for $\gamma = 1$ and $s_{max} = 1$. For comparison, we also plot in Fig. 7 the asymptotic rate ($n \to \infty$) under the assumption that the state of the device is an (unknown) i.i.d. state. In this case, the quantum AEP appearing in Eq. (1) implies that the optimal rate is the von Neumann entropy accumulated in one round of the protocol (as given in Eq. (2)). This rate, appearing as the dashed line in Fig. 7, is an upper bound on the entropy that can be accumulated. One can see that as the number of rounds in the protocol increases, our rate $\eta_{opt}$ approaches this optimal rate.

For the calculations of the DIQKD rates later on, we choose $s_{max} = \lceil \tfrac{1}{\gamma} \rceil$. For this choice, the first-order term of $\eta_{opt}$ is linear in $n$ and a short calculation reveals that the second-order term scales, roughly, as $\sqrt{n/\gamma}$.

Our DIQKD protocol, shown in Box 3, is based on the entropy accumulation protocol described above. In the first part of the protocol Alice and Bob use their devices to produce the raw data, similarly to what is done in the entropy accumulation protocol. The main difference is that Bob's outputs always contains his measurement outcomes (instead of being set to $\perp$ in the generation rounds); to make the distinction explicit, we denote Bob's outputs in the DIQKD protocol with a tilde, $\tilde{B}_1^n$.

We now describe the three post-processing steps, error correction, parameter estimation, and privacy amplification, in more detail.

**Error correction**. Alice and Bob use an error correction protocol EC to obtain identical raw keys $K_A$ and $K_B$ from their bits $A_1^n$, $\tilde{B}_1^n$. In our analysis, we use a protocol, based on universal hashing, which minimises the amount of leakage to the adversary[41,42]. To implement this protocol, Alice chooses a hash function and sends the chosen function and the hashed value of her bits to Bob. We denote this classical communication by $O$. Bob uses $O$, together with his prior knowledge $\tilde{B}_1^n X_1^n Y_1^n T_1^n$, to compute a guess $\hat{A}_1^n$ for Alice's bits $A_1^n$. If EC fails to produce a good guess, Alice and Bob abort; in an honest implementation, this happens with probability at most $\varepsilon_{EC}^c$. If Alice and Bob do not abort, then they hold raw keys $K_A = A_1^n$ and $K_B = \hat{A}_1^n$ and $K_A = K_B$ with probability at least $1 - \varepsilon_{EC}$.

---

**Box 3 | DIQKD protocol (based on the CHSH game)**

**Given:**

 $D$—device that can play the CHSH game repeatedly

 $m \in \mathbb{N}_+$—number of blocks

 $s_{\max} \in \mathbb{N}_+$—maximal length of a block

 $\gamma \in (0, 1]$—probability of a test round

 $\omega_{\exp}$—expected winning probability in the honest implementation

 $\delta_{\mathrm{est}} \in (0, 1)$—width of the statistical confidence interval

 EC—error correction protocol which leaks $\mathrm{leak_{EC}}$ bits and has error probability $\varepsilon_{\mathrm{EC}}$

 PA—privacy amplification protocol with error probability $\varepsilon_{\mathrm{PA}}$

1. For every block $j \in [m]$ do Steps 2–8:
2. Set $i = 0$ and $\tilde{C}_j = \perp$.
3. If $i \le s_{\max}$:
4. Set $i = i + 1$.
5. Alice and Bob choose $T_i \in \{0, 1\}$ at random such that $\Pr(T_i = 1) = \gamma$.
6. If $T_i = 1$ Alice and Bob choose inputs $X_i \in \{0,1\}$ and $Y_i \in \{0,1\}$.
7. If $T_i = 0$ they choose inputs $X_i \in \{0, 1\}$ and $Y_i = 2$.
8. Alice and Bob use $D$ with $X_i$, $Y_i$ and record their outputs as $A_i$, $\tilde{B}_i$.

9. **Error correction:** Alice and Bob apply the error correction protocol EC on the outputs $A_1^n$ and $\tilde{B}_1^n$, communicating $O$ in the process. If EC aborts they abort the protocol. Otherwise, they obtain raw keys denoted by $K_A$ and $K_B$.
10. **Parameter estimation:** Using $\tilde{B}_1^n$ and $K_B$, Bob sets $\tilde{C}_j = w(A_i, B_i, X_i, Y_i)$ for the blocks with a test round at round $i$ and $\tilde{C}_j = \perp$ otherwise. He aborts if $\sum_{j \in [m]} \tilde{C}_j < \left[ \omega_{\exp}(1 - (1 - \gamma)^{s_{\max}}) - \delta_{\mathrm{est}} \right] \cdot m$.
11. **Privacy amplification:** Alice and Bob apply the privacy amplification protocol PA on $K_A$ and $K_B$ to create their final keys $\tilde{K}_A$ and $\tilde{K}_B$ of length $\ell$ as defined in Eq. (9).

---

Due to the communication from Alice to Bob, $\mathrm{leak_{EC}}$ bits of information are leaked to the adversary. The following guarantee holds for the described protocol[42]:

$$\mathrm{leak_{EC}} \le H_0^{\varepsilon'_{\mathrm{EC}}}\left(A_1^n | \tilde{B}_1^n X_1^n Y_1^n T_1^n\right) + \log\left(\frac{1}{\varepsilon_{\mathrm{EC}}}\right), \tag{33}$$

for $\varepsilon_{\mathrm{EC}}^c = \varepsilon'_{\mathrm{EC}} + \varepsilon_{\mathrm{EC}}$ and where $H_0^{\varepsilon'_{\mathrm{EC}}}\left(A_1^n | \tilde{B}_1^n X_1^n Y_1^n T_1^n\right)$ is evaluated on the state in an honest implementation of the protocol. If a larger fraction of errors occur when running the actual DIQKD protocol (for instance due to adversarial interference) the error correction might not succeed, as Bob will not have a sufficient amount of information to obtain a good guess of Alice's bits. If so, this will be detected with probability at least $1 - \varepsilon_{\mathrm{EC}}$ and the protocol will abort. In an honest implementation of the device, Alice and Bob's outputs in the generation rounds should be highly correlated in order to minimise the leakage of information.

**Parameter estimation.** After the error correction step, Bob has all of the relevant information to perform parameter estimation from his data alone, without any further communication with Alice. Using $\tilde{B}_1^n$ and $K_B$, Bob sets $\tilde{C}_j = w_{\mathrm{CHSH}}(\hat{A}_i, \tilde{B}_i, X_i, Y_i) = w_{\mathrm{CHSH}}(K_{Bi}, \tilde{B}_i, X_i, Y_i)$ for the blocks with a test round (which was done at round $i$ of the block) and $\tilde{C}_j = \perp$ otherwise. He aborts if the fraction of successful test rounds is too low, that is, if $\sum_{j \in [m]} \tilde{C}_j < \left[ \omega_{\exp}(1 - (1 - \gamma)^{s_{\max}}) - \delta_{\mathrm{est}} \right] \cdot m$.

As Bob does the estimation using his guess of Alice's bits, the probability of aborting in this step in an honest implementation, $\varepsilon_{\mathrm{PE}}^c$, is bounded by $\varepsilon_{\mathrm{EA}}^c + \varepsilon_{\mathrm{EC}}^c$.

**Privacy amplification.** Finally, Alice and Bob use a (quantum-proof) privacy amplification protocol PA (which takes some random seed $S$ as input) to create their final keys $\tilde{K}_A$ and $\tilde{K}_B$ of length $\ell$, which are close to ideal keys, i.e., uniformly random and independent of the adversary's knowledge.

For simplicity below, we use universal hashing[43] as the privacy amplification protocol in the analysis below. Any other quantum-proof strong extractor, e.g., Trevisan's extractor[44], can be used for this task and the analysis can be easily adapted.

The secrecy of the final key depends only on the privacy amplification protocol used and the value of $H_{\min}^{\varepsilon_s}\left(A_1^n | X_1^n Y_1^n T_1^n OE\right)$, evaluated on the state at the end of the protocol, conditioned on not aborting. For universal hashing, for every $\varepsilon_{\mathrm{PA}}, \varepsilon_s \in (0, 1)$, a secure key of maximal length

$$\ell = H_{\min}^{\varepsilon_s}\left(A_1^n | X_1^n Y_1^n T_1^n OE\right) - 2\log\frac{1}{\varepsilon_{\mathrm{PA}}} \tag{34}$$

is produced with probability at least $1 - \varepsilon_{\mathrm{PA}} - \varepsilon_s$.

Correctness, secrecy, and overall security of a DIQKD protocol are defined as follows[45]:

*Definition 7 (Correctness)*: A DIQKD protocol is said to be $\varepsilon_{\mathrm{corr}}$-correct, when implemented using a device $D$, if Alice and Bob's keys, $\tilde{K}_A$ and $\tilde{K}_B$ respectively, are identical with probability at least $1 - \varepsilon_{\mathrm{corr}}$. That is, $\Pr(\tilde{K}_A \ne \tilde{K}_B) \le \varepsilon_{\mathrm{corr}}$.

*Definition 8 (Secrecy)*: A DIQKD protocol is said to be $\varepsilon_{\mathrm{sec}}$-secret, when implemented using a device $D$, if for a key of length $l$,
$(1 - \Pr[\mathrm{abort}]) \| \rho_{\tilde{K}_A E} - \rho_{U_l} \otimes \rho_E \|_1 \le \varepsilon_{\mathrm{sec}}$, where $E$ is a quantum register that may initially be correlated with $D$.

$\varepsilon_{\mathrm{sec}}$ in the above definition can be understood as the probability that some non-trivial information leaks to the adversary[45]. If a protocol is $\varepsilon_{\mathrm{corr}}$-correct and $\varepsilon_{\mathrm{sec}}$-secret (for a given $D$), then it is $\varepsilon_{\mathrm{QKD}}^s$-correct-and-secret for any $\varepsilon_{\mathrm{QKD}}^s \ge \varepsilon_{\mathrm{corr}} + \varepsilon_{\mathrm{sec}}$.

*Definition 9 (Security)*: A DIQKD protocol is said to be $\left(\varepsilon_{\mathrm{QKD}}^s, \varepsilon_{\mathrm{QKD}}^c, l\right)$-secure if:

1. (Soundness) For any implementation of the device $D$ it is $\varepsilon_{\mathrm{QKD}}^s$-correct-and-secret.
2. (Completeness) There exists an honest implementation of the device $D$ such that the protocol does not abort with probability greater than $1 - \varepsilon_{\mathrm{QKD}}^c$.

Below we show that the following theorem holds.

*Theorem 10:* The DIQKD protocol described above is $\left(\varepsilon_{\mathrm{QKD}}^s, \varepsilon_{\mathrm{QKD}}^c, \ell\right)$-secure, with $\varepsilon_{\mathrm{QKD}}^s \le \varepsilon_{\mathrm{EC}} + \varepsilon_{\mathrm{PA}} + \varepsilon_s + \varepsilon_{\mathrm{EA}}$, $\varepsilon_{\mathrm{QKD}}^c \le \varepsilon_{\mathrm{EC}}^c + \varepsilon_{\mathrm{EA}}^c + \varepsilon_{\mathrm{EC}}$, and

$$\begin{aligned}
\ell = \frac{n}{s} \cdot \eta_{\mathrm{opt}}(\varepsilon_s/4, \varepsilon_{\mathrm{EA}} + \varepsilon_{\mathrm{EC}}) \\
- \mathrm{leak_{EC}} - 3\log\left(1 - \sqrt{1 - (\varepsilon_s/4)^2}\right) - \gamma(n + t) \\
- \sqrt{n + t}\, 2\log 7 \sqrt{1 - 2\log\left((\varepsilon_s/4 - \sqrt{\varepsilon_t}) \cdot (\varepsilon_{\mathrm{EA}} + \varepsilon_{\mathrm{EC}})\right)} \\
- 2\log\left(\varepsilon_{\mathrm{PA}}^{-1}\right),
\end{aligned} \tag{35}$$

where $t = \sqrt{-m(1 - \gamma)^2 \log \varepsilon_t / 2\gamma^2}$ for any $\varepsilon_t \in (0, 1)$.

We now explain the steps taken to prove Theorem 10. The completeness part follows trivially from the completeness of the "subprotocols".

To establish soundness, first note that by definition, as long as the protocol does not abort it produces a key of length $\ell$. Therefore, it remains to verify correctness, which depends on the error correction step, and security, which is based on the privacy amplification step. To prove security we start with Lemma 11, in which we assume that the error correction step is successful. We then use it to prove soundness in Lemma 12.

Let $\tilde{\Omega}$ denote the event of the DIQKD protocol not aborting and the EC protocol being successful, and let $\tilde{\rho}_{A_1^n \tilde{B}_1^n X_1^n Y_1^n T_1^n O_1^n E | \tilde{\Omega}}$ be the state at the end of the protocol, conditioned on this event.

Success of the privacy amplification step relies on the min-entropy $H_{\min}^{\varepsilon_s}\left(A_1^n | X_1^n Y_1^n T_1^n OE\right)_{\tilde{\rho}_{|\tilde{\Omega}}}$ being sufficiently large. The following lemma connects this quantity to $H_{\min}^{\frac{\varepsilon_s}{4}}\left(A_1^n B_1^n | X_1^n Y_1^n T_1^n E\right)_{\rho_{|\tilde{\Omega}}}$, on which a lower bound is provided in Eq. (31) above.

*Lemma 11:* For any device $D$, let $\tilde{\rho}$ be the state generated in the protocol right before the privacy amplification step. Let $\tilde{\rho}_{|\tilde{\Omega}}$ be the state conditioned on not aborting the protocol and success of the EC protocol. Then, for any $\varepsilon_{\mathrm{EA}}, \varepsilon_{\mathrm{EC}}, \varepsilon_s$,

$\varepsilon_t \in (0, 1)$, either the protocol aborts with probability greater than $1 - \varepsilon_{EA} - \varepsilon_{EC}$ or

$$
\begin{aligned}
H_{\min}^{\varepsilon_s}\left(A_1^n | X_1^n Y_1^n T_1^n OE\right)_{\tilde{\rho}_{|\hat{\Omega}}} \geq {} & \frac{n}{s} \cdot \eta_{opt}(\varepsilon_s/4, \varepsilon_{EA} + \varepsilon_{EC}) \\
& - \text{leak}_{EC} - 3 \log\left(1 - \sqrt{1 - (\varepsilon_s/4)^2}\right) \\
& - \gamma(n + t) \\
& - \sqrt{n + t}\, 2 \log 7 \sqrt{1 - 2\log\left((\varepsilon_s/4 - \sqrt{\varepsilon_t}) \cdot (\varepsilon_{EA} + \varepsilon_{EC})\right)}.
\end{aligned}
$$

(36)

*Proof sketch.* Before deriving a bound on the entropy of interest, we remark that the $t$ is chosen such that the probability that the actual number of rounds in the protocol, $N$, is larger than the expected number of rounds $n$ plus $t$ is $\varepsilon_t$. The above value for $t$ can be derived by noticing that the sizes of the blocks are i.i.d. random variables which take values in $[1, 1/\gamma]$.

The key idea of the proof is to consider the following events:

1. $\Omega$: the event of not aborting in the entropy accumulation protocol. This happens when the Bell violation, calculated using Alice and Bob's outputs (and inputs), is sufficiently high.
2. $\hat{\Omega}$: Suppose Alice and Bob run the entropy accumulation protocol, and then execute the EC protocol. The event $\hat{\Omega}$ is defined by $\Omega$ and $K_B = A_1^n$.
3. $\tilde{\Omega}$: the event of not aborting the DIQKD protocol and $K_B = A_1^n$.

The state $\rho_{|\hat{\Omega}}$ then denotes the state at the end of the entropy accumulation protocol conditioned on $\hat{\Omega}$.

Using a sequence of chain rules for smooth entropies[46] and the fact that $\tilde{\rho}_{A_1^n X_1^n Y_1^n T_1^n E | \tilde{\Omega}} = \rho_{A_1^n X_1^n Y_1^n T_1^n E | \hat{\Omega}}$ ($\tilde{B}_1^n$ and $B_1^n$ were traced out from $\tilde{\rho}$ and $\rho$, respectively), one can conclude

$$
\begin{aligned}
H_{\min}^{\varepsilon_s}\left(A_1^n | X_1^n Y_1^n T_1^n OE\right)_{\tilde{\rho}_{|\tilde{\Omega}}} \geq {} & H_{\min}^{\frac{\varepsilon_s}{4}}\left(A_1^n B_1^n | X_1^n Y_1^n T_1^n E\right)_{\rho_{|\hat{\Omega}}} \\
& - H_{\max}^{\frac{\varepsilon_s}{4}}\left(B_1^n | T_1^n E\right)_{\rho_{|\hat{\Omega}}} - \text{leak}_{EC} \\
& - 3 \log\left(1 - \sqrt{1 - (\varepsilon_s/4)^2}\right).
\end{aligned}
$$

(37)

$H_{\max}^{\frac{\varepsilon_s}{4}}\left(B_1^n | T_1^n E\right)_{\rho_{|\hat{\Omega}}}$ can be bounded from above. The intuition is that $B_i \neq \perp$ only when $T_i = 0$, which happens with probability $\gamma$. The exact bound can be calculated using the EAT and is given by

$$
\begin{aligned}
H_{\max}^{\frac{\varepsilon_s}{4}}\left(B_1^n | T_1^n E\right)_{\rho_{|\hat{\Omega}}} < {} & \gamma(n + t) \\
& + \sqrt{n + t}\, 2 \log 7 \sqrt{1 - 2\log\left((\varepsilon_s/4 - \sqrt{\varepsilon_t}) \cdot (\varepsilon_{EA} + \varepsilon_{EC})\right)}.
\end{aligned}
$$

(38)

The above steps together with Eq. (31) conclude the proof.

Using Lemma 11, one can prove that our DIQKD protocol is sound.

*Lemma 12:* For any device $D$ let $\tilde{\rho}$ be the state generated by the DIQKD protocol. Then either the protocol aborts with probability greater than $1 - \varepsilon_{EA} - \varepsilon_{EC}$ or it is $(\varepsilon_{EC} + \varepsilon_{PA} + \varepsilon_s)$-correct-and-secret while producing keys of length $\ell$, as defined in Eq. (35).

*Proof sketch.* Assume the DIQKD protocol did not abort. We consider two cases. First, assume that the EC protocol was not successful (but did not abort). Then Alice and Bob's final keys might not be identical. This happens with probability at most $\varepsilon_{EC}$. Otherwise, assume the EC protocol was successful, i.e., $K_B = A_1^n$. In that case, Alice and Bob's keys must be identical also after the final privacy amplification step.

The secrecy depends only on the privacy amplification step, and for universal hashing a secure key is produced as long as Eq. (34) holds. Hence, a uniform and independent key of length $\ell$ as in Eq. (35) is produced by the privacy amplification step unless the smooth min-entropy is not high enough or the privacy amplification protocol was not successful, which happens with probability at most $\varepsilon_{PA} + \varepsilon_s$.

According to Lemma 11, either the protocol aborts with probability greater than $1 - \varepsilon_{EA} - \varepsilon_{EC}$ or the entropy is sufficiently high to create the secret key.

The expected key rates appearing in Fig. 5 in the main text are given by $r = \ell/n$. The key rate depends on the amount of leakage of information due to the error correction step, which in turn depends on the honest implementation of the protocol as mentioned above. To have an explicit bound, we consider the honest implementation described in the main text. Using Eq. (33) and the AEP, one can

show that the amount of leakage in the error correction step is then given by

$$
\begin{aligned}
\text{leak}_{EC} \leq {} & (n + t) \cdot \left[(1 - \gamma)h(Q) + \gamma h(\omega_{exp})\right] \\
& + \sqrt{n + t}\, 4 \log(2\sqrt{2} + 1) \sqrt{2 \log\left(8/(\varepsilon'_{EC} - 2\sqrt{\varepsilon_t})^2\right)} \\
& + \log\left(8/(\varepsilon'_{EC})^2 + 2/(2 - \varepsilon'_{EC})\right) + \log\left(\frac{1}{\varepsilon_{EC}}\right).
\end{aligned}
$$

(39)

To get the optimal key rates, one should fix the parameters of interest (e.g., $\varepsilon_{QKD}^s$, $\varepsilon_{QKD}^c$, and $n$) and optimise over all other parameters.

**DI randomness expansion.** The entropy accumulation protocol can be used to perform DI randomness expansion as well. In a DI randomness expansion protocol, the honest parties start with a short seed of perfect randomness and use it to create a longer random secret string. For the purposes of randomness expansion, we may assume that the parties are co-located, therefore, the main difference from the DIQKD scheme is that there is no need for error correction (and hence there is no leakage of information due to public communication).

In order to minimise the amount of randomness required to execute the protocol, we adapt the main entropy accumulation protocol by deterministically choosing inputs in the generation rounds $X_i, Y_i \in \{0,1\}$. In particular, there is no use for the input 2 to Bob's device, and no randomness is required for the generation rounds. Aside from the last step of privacy amplification, the remainder of the protocol is essentially the same as the entropy accumulation protocol.

The plotted entropy rates in Fig. 7 are therefore also the ones relevant for a DI randomness expansion.

Since we are concerned here not only with generating randomness but also with expanding the amount of randomness initially available to the users of the protocol, we should also evaluate the total number of random bits that are needed to execute the protocol. Random bits are required to select which rounds are generation rounds, i.e., the random variable $T_1^n$, to select inputs to the devices in the testing rounds, i.e., those for which $T_i = 0$, and to select the seed for the quantum proof extractor used for privacy amplification. All of these can be accounted for using standard techniques and so we omit the detailed explanation and formulas.

**Data availability.** No data sets were generated or analysed during the current study.

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

# ARTICLE

17. Acín, A. et al. Device-independent security of quantum cryptography against collective attacks. *Phys. Rev. Lett.* **98**, 230501 (2007).
18. Masanes, L. Universally composable privacy amplification from causality constraints. *Phys. Rev. Lett.* **102**, 140501 (2009).
19. Pironio, S. et al. Device-independent quantum key distribution secure against collective attacks. *New J. Phys.* **11**, 045021 (2009).
20. Hänggi, E., Renner, R. & Wolf, S. Efficient device-independent quantum key distribution. In *Advances in Cryptology–EUROCRYPT 2010*, 216–234 (Springer, 2010).
21. Masanes, L., Renner, R., Christandl, M., Winter, A. & Barrett, J. Full security of quantum key distribution from no-signaling constraints. *IEEE Trans. Inf. Theory* **60**, 4973–4986 (2014).
22. Bennett, C. H. & Brassard, G. Quantum cryptography: public key distribution and coin tossing. In *Proc. IEEE International Conference on Computers, Systems, and Signal Processing, Bangalore, India*, 175–179 (IEEE, NY, 1984).
23. Renner, R. Symmetry of large physical systems implies independence of subsystems. *Nat. Phys.* **3**, 645–649 (2007).
24. Christandl, M., König, R. & Renner, R. Postselection technique for quantum channels with applications to quantum cryptography. *Phys. Rev. Lett.* **102**, 020504 (2009).
25. Christandl, M. & Toner, B. Finite de Finetti theorem for conditional probability distributions describing physical theories. *J. Math. Phys.* **50**, 042104 (2009).
26. Arnon-Friedman, R. & Renner, R. de Finetti reductions for correlations. *J. Math. Phys.* **56**, 052203 (2015).
27. Reichardt, B. W., Unger, F. & Vazirani, U. Classical command of quantum systems. *Nature* **496**, 456–460 (2013).
28. Vazirani, U. & Vidick, T. Fully device-independent quantum key distribution. *Phys. Rev. Lett.* **113**, 140501 (2014).
29. Miller, C. A. & Shi, Y. Robust protocols for securely expanding randomness and distributing keys using untrusted quantum devices. In *Proc. 46th Annual ACM Symposium on Theory of Computing*, 417–426 (ACM, 2014).
30. Tomamichel, M., Colbeck, R. & Renner, R. Duality between smooth min- and max-entropies. *IEEE Trans. Inf. Theory* **56**, 4674–4681 (2010).
31. Tomamichel, M., Colbeck, R. & Renner, R. A fully quantum asymptotic equipartition property. *IEEE Trans. Inf. Theory* **55**, 5840–5847 (2009).
32. Berta, M., Christandl, M., Colbeck, R., Renes, J. M. & Renner, R. The uncertainty principle in the presence of quantum memory. *Nat. Phys.* **6**, 659–662 (2010).
33. Garcia-Patron, R. & Cerf, N. J. Unconditional optimality of gaussian attacks against continuous-variable quantum key distribution. *Phys. Rev. Lett.* **97**, 190503 (2006).
34. Clauser, J. F., Horne, M. A., Shimony, A. & Holt, R. A. Proposed experiment to test local hidden-variable theories. *Phys. Rev. Lett.* **23**, 880 (1969).
35. Scarani, V. & Renner, R. Quantum cryptography with finite resources: unconditional security bound for discrete-variable protocols with one-way postprocessing. *Phys. Rev. Lett.* **100**, 200501 (2008).
36. Wilde, M. M., Winter, A. & Yang, D. Strong converse for the classical capacity of entanglement-breaking and hadamard channels via a sandwiched rényi relative entropy. *Commun. Math. Phys.* **331**, 593–622 (2014).
37. Müller-Lennert, M., Dupuis, F., Szehr, O., Fehr, S. & Tomamichel, M. On quantum rényi entropies: a new generalization and some properties. *J. Math. Phys.* **54**, 122203 (2013).
38. Tomamichel, M. A framework for non-asymptotic quantum information theory. Preprint at https://arxiv.org/abs/1203.2142 (2012).
39. Müller-Lennert, M. *Quantum relative Rényi entropies*. Master's thesis (ETH Zürich, 2013).
40. Inequalities for the moments of the eigenvalues of the Schrödinger equation and their relations to Sobolev inequalities. In *Studies in Mathematical Physics: Essays in honor of Valentine Bargman*, 269–303 (1976).
41. Brassard, G. & Salvail, L. Secret-key reconciliation by public discussion. In *Advances in Cryptology EUROCRYPT 93*, 410–423 (Springer, 1993).
42. Renner, R. & Wolf, S. Simple and tight bounds for information reconciliation and privacy amplification. In *Advances in cryptology-ASIACRYPT 2005*, 199–216 (Springer, 2005).
43. Renner, R. & König, R. Universally composable privacy amplification against quantum adversaries. In *Theory of Cryptography*, 407–425 (Springer, 2005).
44. De, A., Portmann, C., Vidick, T. & Renner, R. Trevisan's extractor in the presence of quantum side information. *SIAM J. Comput.* **41**, 915–940 (2012).
45. Portmann, C. & Renner, R. Cryptographic security of quantum key distribution. Preprint at https://arxiv.org/abs/1409.3525 (2014).
46. Tomamichel, M. *Quantum Information Processing with Finite Resources: Mathematical Foundations*, Vol. 5 (Springer, 2015).

## Acknowledgements

We thank Asher Arnon for the illustrations presented in Figs. 1, 4, and 5. R.A.F. and R.R. were supported by the Stellenbosch Institute for Advanced Study (STIAS), by the European Commission via the project "RAQUEL", by the Swiss National Science Foundation (grant No. 200020–135048) and the National Centre of Competence in Research "Quantum Science and Technology", by the European Research Council (grant No. 258932), and by the US Air Force Office of Scientific Research (grant No. FA9550-16-1-0245). F.D. acknowledges the financial support of the Czech Science Foundation (GA ČR) project no GA16-22211S and of the European Commission FP7 Project RAQUEL (grant No. 323970). O.F. acknowledges support from the LABEX MILYON (ANR-10-LABX-0070) of Université de Lyon, within the program "Investissements d'Avenir" (ANR-11-IDEX-0007) operated by the French National Research Agency (ANR). T.V. was partially supported by NSF CAREER Grant CCF-1553477, an AFOSR YIP award, the IQIM, and NSF Physics Frontiers Center (NFS Grant PHY-1125565) with support of the Gordon and Betty Moore Foundation (GBMF-12500028).

## Author contributions

R.A.F., F.D., O.F., R.R., and T.V. contributed equally to this work.

## Additional information

**Competing interests:** The authors declare that they have no competing financial interests.

