## [Peer Review File · Nature Communications]

REVIEWERS' COMMENTS:

Reviewer #1 (Remarks to the Author):

Report on Practical device-independent quantum cryptography

The authors considerably advance the field of device independent quantum crypto, by showing that entropy bounds obtained using an iid hypothesis are comparable to those not making this assumption. This is an important advance, that makes the whole field much closer to practical relevance. I recommend the paper for publication. I have a few very minor suggestions.

Minor comments:

Main text.

-In the intro the authors could mention the application of DI approach to computation (they cite, but later, a key paper (ref 29, Reichardt et al)).

-Description of the EAT and Fig .5. The reason why systems O_i and S_i need to be considered separately is not clearly indicated.

-Conclusion. Saying that DI qkd is within reach of present day technology is pushing things a bit. Achieving reasonable distances and key rates is a major challenge, without a clear route forward to achieve them. Maybe change slightly the formulation.

Supplementary Info.

Definition 1 (EAT channel). Point 1. Dimension of O_i is d_{O_i} . Is any hypothesis needed on d_{O_i} ?

Fig 11 DIQKD protocol. Step 5. Choosing T_i requires randomness. If this is secret shared randomness, then it should be stated, and taken into account.

Reviewer #2 (Remarks to the Author):

This is an excellent paper and, in my opinion, should be published in its current form (the authors may consider few minor remarks that I have appended).

To start with, the authors prove a powerful mathematical statement about entropy, which is a generalisation of the quantum asymptotic equipartition property to cases which are not of the independent and identically distributed (IID) type. The beauty of this result, which the authors call “entropy accumulation theorem” (EAT), is that even under weaker assumptions (sequential quantum process) entropy accumulations is essentially proportional to the number of rounds (in a CHSH game played on the system) with the conditional von Neumann entropy being the relevant single-round quantity. This leads to a neat security proof of the device-independent quantum key distribution, essentially showing that an adversary can hardly do better than in the IID scenario. Once this is shown, the whole set of well established results pertaining to the IID case can be brought in and used - the quantum crypto community will find it both reassuring and comforting. This said, one would expect such a nice result to come with a simple intuitive justification or at least a good plausibility argument. This, unfortunately, is not provided and one is left digesting lines of mathematics in the supplementary section, drudgery of which is alleviated by a very clear exposition. Both the main and the supplementary sections are well written and all the mathematical concepts and techniques are carefully explained. It is clear that the authors have made a substantial effort to make this paper accessible to a broad audience of researchers, in particular physicists and cryptologists.

The EAT is such an important and powerful tool that it is bound to have many interesting applications and, almost certainly, will inspire new research directions. The resulting proof of the security of the device-independent quantum key distribution indicates that the device independent crypto may be within the experimental reach in a not too distant future. Thus, as far as I am concerned, the paper ticks all the boxes in favour of publication - important, original and broad interest.

Here are few minor / optional suggestions

- Fig. 2, caption. Wording: consider “Secrecy for the CHSH game vs. winning probability” instead of “Winning probability vs. secrecy for the CHSH game”. Mention that the plot is based on Eq. 2 and only then refer to (18). The relation is indeed derived in the cited paper but the plots in (18) have different parameterisation and one will find nothing there that resembles Fig. 2.

- Right after Eq. 1 (...where $c\epsilon$ depends only on ϵ). You may want to reassure the readers that this dependence is not, say, of exponential type (refer to the supplementary section or just comment on this dependence). This becomes obvious later on, but some may read this paper in a “sequential way” (pun intended).

- Right before Eq. 2. Consider shifting “where $h(\cdot)$ the binary entropy function” from right before to right after Eq. 2.

- Fig.5, caption. The last sentence about the final state may require some explanation (tracing over R_n) if it were to go with the diagram.

- Fig. 6, caption. It would be helpful to mention which formulae were used for the plots and refer to the completeness and the soundness errors also by their symbols (as used in the supplementary material).

Regards,

Artur Ekert

Reviewer #3 (Remarks to the Author):

In this paper the authors present a new technical result called the Entropy Accumulation Theorem (EAT) and use it to present a proof of correctness and security of sequential Device Independent Quantum Key Distribution (DIQKD). Earlier proofs of sequential DIQKD assumed an i.i.d setting which the current proof does not. This is a major improvement as far as practical implementations of QKD are concerned.

The proof of EAT is information theoretic and quite involved. EAT is likely to be an important fact of independent importance and interest in the future. EAT basically states that entropy (even given side information) accumulates (under suitable conditions) for various random variables obtained as a result of sequential execution of various quantum maps, even when memory is passed from previous maps to subsequent maps.

The authors claim that important parameters of their DIQKD protocol e.g. error tolerance and number of signals has been significantly improved compared to protocols from previous works. Please mention here the best parameters of previous works and how much the current work improves upon them.

There are some recent works where parallel (as opposed to sequential) implementations of DIQKD have been analysed and correctness and security proofs given (e.g. arXiv:1707.06597, arXiv:1703.05426, arXiv:1703.08508.) It seems that sequential DIQKD is implied by parallel DIQKD. Is this correct. If yes, is the current work doing better in terms of some significant parameters? Please elaborate on comparison. Also it seems that the proofs of parallel DIQKD are significantly simpler than the proofs in the current work. Is that so? Please compare in this regards.

It is important to receive the inputs as mentioned above from the authors before a recommendation (regarding acceptance/rejection) can be made.

Reply to referees

We would like to thank the referees for their careful reading of our manuscript and for all of their helpful suggestions, comments, and questions. Our replies are given below.

Reviewer #1 (Remarks to the Author):

Report on Practical device-independent quantum cryptography

The authors considerably advance the field of device independent quantum crypto, by showing that entropy bounds obtained using an iid hypothesis are comparable to those not making this assumption. This is an important advance, that makes the whole field much closer to practical relevance. I recommend the paper for publication. I have a few very minor suggestions.

Minor comments:

Main text.

-In the intro the authors could mention the application of DI approach to computation (they cite, but later, a key paper (ref 29, Reichardt et al)).

Reply:

The DI approach and the tools developed while studying DI cryptography are also used in other applications and research questions, such as delegated computation (as the referee mentions), complexity theory (in multi-prover interactive proof systems, for example), and DI entanglement certification.

While these applications are clearly important, they are not directly related to our work. Nature Communication limits the length of the introduction of the manuscript and therefore, unfortunately, not all works which use the general DI approach can be mentioned. Since we do not think that mentioning these works in the introduction helps the reader understand our work we chose not to add it at the price of deleting something else.

As a follow-up to the referee's remark, we would like to add that we believe it is interesting to ask whether our techniques can be used in the context of delegated computation. Looking at the structure of the known proofs for delegated computation, it is not clear if this is indeed the case.

-Description of the EAT and Fig .5. The reason why systems O_i and S_i need to be considered separately is not clearly indicated.

Reply:

We clarified this both in the main text and in the figure. The reason to consider them separately is that O_i describes the information which one would like to keep secret while S_i describes the side-information that is leaked by the map M_i . Indeed, the relevant quantity to bound in the end is the smooth min-entropy of the O systems given the S systems and E .

-Conclusion. Saying that DI qkd is within reach of present day technology is pushing things a bit. Achieving reasonable distances and key rates is a major challenge, without a clear route forward to achieve them. Maybe change slightly the formulation.

Reply:

We agree with the reviewer that, while DIQKD is "within reach" (which means it is a reasonable project for an experimentalist to start now), the distances and rates will most likely be very small. We modified the conclusion to say that the first proof of principle experiments, with small distances and small rates, are within reach with today's state-of-the-art technology. We believe this statement is true (see for example a recent experimental work presented in arXiv:1709.06779, where a positive smooth min-entropy rate was certified using our techniques).

Supplementary Info.

Definition 1 (EAT channel). Point 1. Dimension of O_i is d_{O_i} . Is any hypothesis needed on d_{O_i} ?

Reply:

d_{O_i} appears later in Theorem 3. Some bound on d_{O_i} is therefore necessary in order to get a final quantitative result. Note that in the DI setting d_{O_i} describes the dimension of the classical outputs (i.e., the number of possible outputs) and, hence, it is known (in contrast to the dimension of the quantum states which is, of course, unknown).

Fig 11 DIQKD protocol. Step 5. Choosing T_i requires randomness. If this is secret shared randomness, then it should be stated, and taken into account.

Reply:

T_i is generated using public randomness. In other words, one can think of a scenario in which Alice randomly chooses T_i and then communicates its value to Bob using a public (but authenticated) classical channel. (Such a channel is used in all QKD protocols because there are always steps requiring classical communication). This means that the values of T_i are known to the adversary (once they are produced and communicated, not before) and, hence, should be considered as part of the adversary's side information when calculating the lower bound on the conditional smooth min-entropy. This information is, indeed, taken into account in our proof, as can be seen, for example, in Lemma 11.

Reviewer #2 (Remarks to the Author):

This is an excellent paper and, in my opinion, should be published in its current form (the authors may consider few minor remarks that I have appended).

To start with, the authors prove a powerful mathematical statement about entropy, which is a generalisation of the quantum asymptotic equipartition property to cases which are not of the independent and identically distributed (IID) type. The beauty of this result, which the authors call “entropy accumulation theorem” (EAT), is that even under weaker assumptions (sequential quantum process) entropy accumulation is essentially proportional to the number of rounds (in a CHSH game played on the system) with the conditional von Neumann entropy being the relevant single-round quantity. This leads to a neat security proof of the device-independent quantum key distribution, essentially showing that an adversary can hardly do better than in the IID scenario. Once this is shown, the whole set of well established results pertaining to the IID case can be brought in and used - the quantum crypto community will find it both reassuring and comforting.

This said, one would expect such a nice result to come with a simple intuitive justification or at least a good plausibility argument. This, unfortunately, is not provided and one is left digesting lines of mathematics in the supplementary section, drudgery of which is alleviated by a very clear exposition.

Reply:

We agree with the referee that the formal statement of the EAT and its proof are hard to digest in a first read. We added explanations which, hopefully, will help the reader gain some intuition. These appear between Definition 2 and Theorem 3 (now in the Methods section).

Both the main and the supplementary sections are well written and all the mathematical concepts and techniques are carefully explained. It is clear that the authors have made a substantial effort to make this paper accessible to a broad audience of researchers, in particular physicists and cryptologists.

The EAT is such an important and powerful tool that it is bound to have many interesting applications and, almost certainly, will inspire new research directions. The resulting proof of the security of the device-independent quantum key distribution indicates that the device independent crypto may be within the experimental reach in a not too distant future. Thus, as far as I am concerned, the paper ticks all the boxes in favour of publication - important, original and broad interest.

Here are few minor / optional suggestions

- Fig. 2, caption. Wording: consider “Secrecy for the CHSH game vs. winning probability” instead of “Winning probability vs. secrecy for the CHSH game”. Mention that the plot is based on Eq. 2 and only then refer to (18). The relation is indeed derived in the cited paper but the plots in (18) have different parameterisation and one will find nothing there that resembles Fig. 2.
- Right after Eq. 1 (...where ϵ depends only on ϵ). You may want to reassure the readers that this dependence is not, say, of exponential type (refer to the supplementary section or just comment on this dependence). This becomes obvious later on, but some may read this paper in a “sequential way” (pun intended).
- Right before Eq. 2. Consider shifting “where $h(\cdot)$ the binary entropy function” from right before to right after Eq. 2.
- Fig.5, caption. The last sentence about the final state may require some explanation (tracing over R_n) if it were to go with the diagram.
- Fig. 6, caption. It would be helpful to mention which formulae were used for the plots and refer to the completeness and the soundness errors also by their symbols (as used in the supplementary material).

Reply:

We followed all of the referee’s suggestions.

Reviewer #3 (Remarks to the Author):

In this paper the authors present a new technical result called the Entropy Accumulation Theorem (EAT) and use it to present a proof of correctness and security of sequential Device Independent Quantum Key Distribution (DIQKD). Earlier proofs of sequential DIQKD assumed an i.i.d setting which the current proof does not. This is a major improvement as far as practical implementations of QKD are concerned.

The proof of EAT is information theoretic and quite involved. EAT is likely to be an important fact of independent importance and interest in the future. EAT basically states that entropy (even given side information) accumulates (under suitable conditions) for various random variables obtained as a result of sequential execution of various quantum maps, even when memory is passed from previous maps to subsequent maps.

The authors claim that important parameters of their DIQKD protocol e.g. error tolerance and number of signals has been significantly improved compared to protocols from previous works. Please mention here the best parameters of previous works and how much the current work improves upon them.

Reply:

This is clarified in the revised version (in the last paragraph before the conclusion of the manuscript). We further remark that none of the previous security proofs for DIQKD gave explicit bounds for finite number of signals. However, as we describe in the paper, even when comparing the previous explicit asymptotic bounds to our finite statistic bounds, our key rates are significantly better.

There are some recent works where parallel (as opposed to sequential) implementations of DIQKD have been analysed and correctness and security proofs given (e.g. arXiv:1707.06597, arXiv:1703.05426, arXiv:1703.08508.) It seems that sequential DIQKD is implied by parallel DIQKD. Is this correct.

Reply:

The mentioned works on executing protocols for DIQKD in parallel all appeared subsequently to our work. They are interesting theoretical works, but in our opinion they focus on a wholly different aspect of DIQKD than we do.

The two settings, sequential and parallel, are incomparable. The reason is that when considering the parallel setting (in which all the Bell games are being played together) one does not allow for communication between the rounds of the protocol: it is assumed that the devices are provided with their respective n inputs “simultaneously”, and that they also provide outputs simultaneously. In contrast our proof allows communication in-between rounds. We remark it is necessary to consider some form of communication between the rounds if one wishes to implement an experiment in which the distribution of the quantum states is done “on the fly” (in contrast to providing all n entangled pairs at the same time).

In our understanding the references you cite focus on exploring a different setting for DIQKD (as discussed above), and in getting proofs for that setting, by showing that it can be analysed using techniques from parallel repetition and other ideas. We believe that considering parallel DIQKD is conceptually interesting since it requires a security proof with a structure that is very different from ours (or of other works analysing sequential DIQKD). However, it is unclear whether they are relevant to experiments of the form considered today, which use devices that can only process one signal at a time and are hence inherently sequential.

If yes, is the current work doing better in terms of some significant parameters? Please elaborate on comparison.

Reply:

Although the answer to the previous question is 'no', one may compare the parameters. Neither works on parallel DIQKD provides any advantage in terms of parameters compared to sequential DIQKD. Quantitatively, the key rates presented in our work are significantly better in all parameters; we believe that sequential DIQKD will maintain a very strong advantage (partially due to our proof technique) in this respect for the foreseeable future.

Also it seems that the proofs of parallel DIQKD are significantly simpler than the proofs in the current work. Is that so? Please compare in this regards.

Reply:

We disagree with this assessment. The works on parallel DIQKD are based on results on quantum parallel repetitions, which are highly involved themselves. Moreover, note that the current proofs for parallel DIQKD are merely "proofs of concept" as they require parameter choices that are not achievable in realistic settings. The hard part of the work will be to find new techniques which can be used to derive stronger and realistic key rates.

We hope that the above replies clarify the crucial differences between the parallel and sequential settings as well as the works themselves.

It is important to receive the inputs as mentioned above from the authors before a recommendation (regarding acceptance/rejection) can be made.

Corresponding author: Rotem Arnon-Friedman

Rotem Arnon-Friedman